# COVID-19 Pandemic Increases the Risk of Anxiety and Depression among Adolescents: A Cross-Sectional Study in Rural West Bengal, India

**DOI:** 10.3390/diseases12100233

**Published:** 2024-10-01

**Authors:** Amal K. Mitra, Sinjita Dutta, Aparajita Mondal, Mamunur Rashid

**Affiliations:** 1Department of Public Health, Julia Jones Matthews School of Population and Public Health, Texas Tech University Health Sciences Center, Abilene, TX 79601, USA; 2Department of Community Medicine, Institute of Post-Graduate Medical Education and Research, 240 AJC Bose Road, Kolkata 700020, West Bengal, India; sinjita@gmail.com (S.D.); aparajitamondal06@gmail.com (A.M.); 1998mamunur@gmail.com (M.R.)

**Keywords:** prevalence, mental health, depression, anxiety, adolescents, COVID-19, pandemic

## Abstract

About 14% of adolescents experience mental illnesses globally. The rate increased after the COVID-19 pandemic. This study aimed to estimate the prevalence of COVID-related major mental illnesses (depression and anxiety) and their predictors among adolescents. This community-based cross-sectional study was conducted among 350 adolescents aged 10–19 years, recruited from rural communities in West Bengal, India. The study areas included 27 subcenters in the Budge Budge II Block, which comprised 191,709 population and 45,333 households. Data were collected from randomly selected households by trained research assistants through house-to-house interviews. Pretested standardized questionnaires, Patient Health Questionnaire (PHQ) for depression, Generalized Anxiety Disorder (GAD) for anxiety, and a demographic questionnaire were used for data collection. The prevalence of anxiety was 35.7% (125 out of 350), and the prevalence of depression was 30.0% (105 out of 350). Females had significantly higher scores of anxiety (6.67 ± 4.76 vs. 3.42 ± 3.17, respectively, *p* < 0.001) and depression (5.51 ± 4.99 vs. 3.26 ± 3.86, respectively, *p* < 0.001) compared to males. Adolescents who had COVID-19 cases or deaths in the family had significantly higher scores of anxiety and depression compared to those who did not have these events. In multivariate analysis, the statistically significant predictors of anxiety were COVID-19 death in the family, COVID-19 cases in the family, female gender, and a lower income group (*p* < 0.001 for all). For depression, the significant predictors included COVID-19 death in the family, COVID-19 cases in the family, female gender, a lower income group, and a higher age group in adolescents (*p* < 0.001 for all). Based on the study results, we recommend that immediate attention is needed for adolescents’ mental health support and coping with stresses following COVID-19.

## 1. Introduction

Adolescence is a critical developmental period, yet globally, an estimated 166 million adolescents experience mental illnesses [1]. Major mental illnesses include anxiety, depression, and health risk behaviors, such as illicit drug use, smoking, drinking, and suicidality. While recent data from the United States show a decreasing trend in drug use among adolescents [2], this pattern is not universal, highlighting the complex nature of adolescent mental health. Consequently, adolescents are often victims of negative mental health outcomes, developmental issues, and threats to brain functions, resulting in short- and long-term mental illnesses [3,4].

The COVID-19 pandemic has significantly impacted mental health worldwide. Factors such as fear of infection, resource insecurity, and social isolation have contributed to increased mental health issues [5,6]. The United States Surgeon General has warned of a potential public health crisis due to the pandemic’s impact on mental health [7]. India, with the second-highest burden of COVID-19 cases globally [6], faces particular challenges in this regard.

As a result of social isolation and the continued fear of infection and death due to the virus, negative mental health consequences that are associated with the COVID-19 crisis are monumental [8,9]. The psychological toll of COVID-19 on adolescents has received less attention compared to other age groups [10,11], despite the unique vulnerabilities of this population. School closures, quarantine periods, lack of recreational activities, social distancing, loss of loved ones, and loneliness have exacerbated threats to adolescent mental health [12,13]. Adolescents’ limited psychological maturity, coping techniques, and mental resilience compared to adults further compound these challenges [14,15].

In India, the situation is particularly concerning due to persistent inequalities based on socioeconomic status, rural-urban divides, and gender differences in health care and health status in several parts of the country [16]. This is compounded by the lack of healthcare access for most socioeconomically disadvantaged people in the rural communities of India. A recent study in India [17] suggested that the pandemic may increase risks of school dropout, gender gaps in education, child labor, stress, smartphone addiction, substance use initiation, and exposure to violence, neglect, or exploitation among adolescents. Another review study emphasized the need for evidence-based action plans addressing the psychosocial and mental health needs of vulnerable children [18].

However, most studies on COVID-19’s impact on adolescent mental health have focused on urban populations in developed countries or relied on review articles and retrospective data analysis [11,12,13,14,17,18]. Data on the pandemic’s effect on adolescent mental health in rural or underprivileged areas, particularly in India, are scarce. One online cross-sectional survey by Dev et al. (2023) [19] explored this topic, but its focus on an older age group (16–29 years) and use of different measurement scales (e.g., the Perceived Stress Scale, Adult Hope Scale, and Mental Health Inventory) limit comparability with our study.

Given this context, there is a pressing need to address the knowledge gap regarding the prevalence and risk factors of mental health issues among adolescents in underprivileged rural communities during the COVID-19 pandemic. This information is crucial for developing strategies to prevent poor mental health outcomes in these populations.

We hypothesize that anxiety and depression are more prevalent among adolescents from families with COVID-19 cases and/or deaths compared to those without such experiences. Our primary objective is to identify the prevalence of anxiety and depression in adolescents among families with or without COVID-19 cases and deaths. Our secondary objective is to identify risk factors for anxiety and depression, controlling for demographic factors.

This study has broader global implications, as its findings from a rural population can inform public health policies and interventions in similar settings worldwide.

## 2. Materials and Methods

### 2.1. Study Population and Study Design

A cross-sectional community-based study was conducted among adolescents residing under the aegis of the 27 subcenters of Budge-Budge II Block, West Bengal from January–June 2023. The Institute of Post-Graduate Medical Education and Research (IPGME&R), Kolkata, India had ongoing research and service centers (also called rural field practice areas) in *Muchisa* (Figure 1). These rural field practice areas are located in the Budge Budge II Block of the South 24 Parganas district of West Bengal. IPGME&R maintained services in 27 subcenters and in one rural hospital. A list of all households was also available for these rural areas. Due to the established rapport with the rural people, we selected the rural community of *Muchisa* as the study population for the current research. The study areas are shown in Figure 1.

### 2.2. Inclusion and Exclusion Criteria

Adolescents between 10–19 years of age and those residing in the study area for more than 3 years were included in this study. Those participants previously having any mental health disorders either self-reported or diagnosed by a doctor, or with any ongoing major health issues (such as congenital disease, other comorbidities—pneumonia, asthma, heart disease, etc.) or were absent from their homes during the data collection period on three separate occasions were excluded from the study.

### 2.3. Sample Size Estimation

The major variable of interest in the study was the prevalence of depression. We used the study reported by Shah and Bhattad (2022), in which the prevalence of depression was 13% among patients following COVID-19 [4]. We used the following sample size formula for estimating the expected prevalence of depression [20]. Data for estimating anxiety was not available from the study of Shah and Bhattad used here. However, most other studies showed a higher prevalence of anxiety. Therefore, the estimated sample size for anxiety should be lower than that for depression.
$$n=\frac{Z^{2}pq}{d^{2}}$$

Here, the proportion of the people with depression, *p* = 13% or 0.13

*q* = 1 − *p* = 0.87

*Z* = 1.96 for *α* = 0.05 (having 95% confidence)

Absolute precision, *d* = 4% or 0.04
$$n=\frac{({1.96)}^{2}\times 0.13\times 0.87}{{\left(0.04\right)}^{2}}\;=272$$

Adjusting for a non-response rate of 15%, the minimum required sample size was 313; while we collected data from 350 eligible adolescents.

### 2.4. Sampling Technique

The sampling frame was obtained from the line list of households with adolescents maintained at the subcenters. Each subcenter was considered a cluster. First, we determined the number of clusters (or subcenters) required to include the required 350 samples out of the total number of adolescents in all clusters. For the second step, the required clusters were selected randomly from the total number of clusters (or subcenters). The third step was to approach each household from the line list of households available in the cluster sample. Finally, adolescents from each cluster (or subcenter) were selected by simple random sampling without replacement until the desired number of adolescents (*n* = 350) was included.

Several potential biases were anticipated in the sample selection process; they include the following: (1) People’s cooperation—IPGME&R and the SSKMH Hospital, Kolkata, West Bengal being a highly reputed tertiary care hospital in West Bengal, we did not encounter this problem of selection bias because of people’s noncooperation. (2) Attrition bias—the people in these subcenters were generally a stable population. In case an adult household member was absent during our visit, we attempted three times to reach the household and then moved to the next household in the list. In fact, there was no dropout of the sample once enrolled.

Accredited Social Health Activists (*ASHA*) from the respective areas helped identify the selected households. Two trained research assistants from IPGME&R visited randomly selected households (Figure 2).

### 2.5. Ethical Considerations

The study was conducted following the Declaration of Helsinki, and the protocol was approved by the Institutional Ethics Committee (IEC) of IPGME&R/SSKMH hospital (letter number IPGME&R/IEC/2023/047, dated 21 January 2023). Informed written consents and assents (above age 7) were taken before enrollment. The study objectives and the procedures were explained before the informed consent was obtained. Participation in the study was completely voluntary, and the participants had the right to withdraw from the study at any time during the study. They were assured about the anonymity, confidentiality of the data, and risks and benefits. No body samples (such as blood or urine) were obtained.

### 2.6. Study Tools and Techniques

A predesigned, pretested, structured data instrument was used to collect data. The data instrument included the following: Demographic data, a COVID-related questionnaire, and two standardized questionnaires: (a) Patient Health Questionnaire (PHQ)-9, and (b) Generalized Anxiety Disorder (GAD)-7 scales [21,22]. PHQ is a nine-item screening method that offers concise, self-administered tools for assessing depression. Participants will choose one of four responses about the frequency of depression during the previous two weeks. PHQ-9 scores > 10 have a sensitivity of 0.88 (95% confidence intervals, 0.82 to 0.92) and a specificity of 0.86 (95% CI, 0.82 to 0.85) for major depressive disorder [23]. GAD-7 has a sensitivity of 0.83 (95% CI, 0.71 to 0.91) and a specificity of 0.84 (95% CI, 0.70 to 0.92) for screening anxiety disorders [24]. Both questionnaires have been extensively used around the world to assess mental health status. All questionnaires, including PHQ-9 and GAD-7, were translated from English to the local language (Bengali), and back-translated, as 98% of the study population spoke Bengali. During the process, two questions (Q6 and Q8 of PHQ) were articulated differently in Bengali from the intended inherent meanings of the original questions in English. These questions were rephrased and pre-tested. The final pre-tested Bengali version of the instruments were incorporated for data collection.

The local communities were approached, and households were identified by ASHA workers in the area. Trained interviewers made house-to-house visits to the selected households to administer the questionnaire. Data collection was monitored by the researchers to ensure quality.

### 2.7. Statistical Analysis

Data were entered, cleaned, and analyzed using SPSS (version 29, Armonk, N.Y.: IBM Corp.). We assessed the data distribution by conventional methods, such as descriptive analysis, histogram, Boxplot, Q-Q plot, normality plot, and Shapiro-Wilk Test of Normality. The numerical values of monthly family income were categorized into four income groups. The total numerical depression scores were categorized as follows: none-to-minimal (scores 0–4); mild (scores 5–9); moderate (scores 10–14), moderately severe (scores 15–19); and severe (scores 20–27) [21]. Whereas the total numerical anxiety scores were subdivided into the following categories: minimal (scores 0–4); mild (scores 5–9); moderate (scores 11–14); and severe (scores 15 and above) [22]. Chi-square test was used to compare the anxiety and the depression categories by gender. Data of the total scores being skewed, median (interquartile range) and a Mann-Whitney U test were computed to find statistical differences of the total scores of anxiety and depression between males and females. A bivariate correlation test was computed between depression and anxiety numerical scores with the demographic variables. To identify significant predictor variables for the total numerical scores of anxiety and depression, after controlling for demographic variables (such as age, gender, and income), and COVID-19 cases and deaths in the family, multiple linear regression analyzes were computed for the dependent variables, anxiety and depression scores, separately. A *p*-value of ≤ 0.05 was used for statistical significance.

## 3. Results

### 3.1. COVID-19 and General Health

The study included 350 adolescents and had no dropouts. As mentioned in Table 1, the mean ± *SD* age of the participants was 15.56 ± 1.93. As expected, most of the families were from lower economic groups, because the study was conducted in rural areas of West Bengal, India.

Among the adolescents, 57 (16.3%) had occurrences of COVID-19 and 8 (2.3%) had COVID-19 deaths in the family. Twenty-seven of the adolescents reported having current illnesses at the time of the interview. Of those who had symptoms (*n* = 27), the majority (59.3%) asked pharmacy sellers for their medication and four (14.8%) used self-medication. Most of the symptoms were related to upper respiratory infections such as sore throat and cough but no fever. Only five (1.4%) adolescents reported having COVID-19. Only three (0.9%) perceived that they might have anxiety or depression. When asked about their overall health, the majority (88%) rated their health “very good” and about 10% mentioned it “good or moderate” (Table 1).

### 3.2. Prevalence of Anxiety and Depression

Table 2 shows that 125 (35.7%) of the adolescents had anxiety. Of the four categories of severity scores (minimal, mild, moderate, and severe), females had significantly higher severity of anxiety than males (*p* < 0.001).

Table 3 depicts the data on depression by gender. A total of 105 (30%) adolescents had depression. Of the severity categories, 97 (27.7%) adolescents had mild or moderate depression, while only eight (2.3%) had moderately severe or severe depression. Among these categories, females had significantly higher severity of depression than males (*p* < 0.001).

Because of the non-normal distribution, we used Mann-Whitney U Tests to determine the gender differences in the anxiety and depression scores. Table 4 shows that the median (interquartile range) scores of anxiety and depression were significantly higher in females than males (*p* < 0.001 for both).

### 3.3. Association between COVID-19 Cases in the Family with Anxiety and Depression

Mean ± *SE* scores of anxiety and depression were compared between adolescents who had COVID-19 cases in the family (*n* = 57) and those who did not (*n* = 293). For anxiety, those who had COVID-19 cases in the family showed significantly higher GAD-7 scores compared to those who did not have COVID-19 cases in the family (6.79 ± 2.59 vs. 4.62 ± 4.50, respectively, Mann -Whitney U *p*-value < 0.001) (Figure 3).

For depression, those who had COVID-19 cases in the family showed significantly higher PHQ-9 scores compared with those who did not have COVID-19 cases in the family (8.47 ± 2.41 vs. 3.54 ± 4.47, respectively, Mann-Whitney U *p*-value < 0.001) (Figure 3).

### 3.4. Association between COVID-19 Death in the Family with Anxiety and Depression

Mean ± *SE* scores of anxiety and depression were compared between adolescents who had COVID-19 death in the family (*n* = 8) and those who did not (*n* = 342). For anxiety, those who had COVID-19 death in the family showed a significantly higher GAD-7 scores compared with those who did not have COVID-19 death in the family (15.63 ± 1.06 vs. 4.73 ± 4.05, respectively, Mann-Whitney U *p*-value < 0.001) (Figure 4).

For depression, those who had COVID-19 death in the family showed significantly higher PHQ-9 scores compared with those who did not have COVID-19 death in the family (17.25 ± 1.39 vs. 4.04 ± 4.17, respectively, Mann-Whitney U *p*-value < 0.001) (Figure 4).

### 3.5. Association between Anxiety and Depression with Demographic Variables and COVID-19

Table 5 shows a bivariate correlation between age, gender, income group, COVID-19 cases in the family, COVID-19 death in the family, anxiety, and depression. The major outcome variables of this included anxiety and depression. Anxiety was significantly correlated with the following variables: age (*r* = 0.123, *p* = 0.022), meaning anxiety increased with increasing age; more in females than males (1 = male, 2 = female) (*r* = 0.376, *p* < 0.001); more in lower income group (*r* = −0.292, *p* < 0.001); having COVID-19 cases (1 = yes, 2 = no) in the family (*r* = −0.185, *p* < 0.001); and having COVID-19 death (1 = yes, 2 = no) in the family (*r* = −0.377, *p* < 0.001).

Depression was significantly correlated with the following variables: more in females than males (1 = male, 2 = female) (*r* = 0.0.246, *p* < 0.001); more in lower income group (*r* = −0.272, *p* < 0.001); having COVID-19 cases (1 = yes, 2 = no) in the family (*r* = −0.399, *p* < 0.001); and having COVID-19 death (1 = yes, 2 = no) in the family (*r* = −0.432, *p* < 0.001).

### 3.6. Multivariate Analysis of the Predictors of Anxiety

Multivariate linear regression analyzes were conducted using the stepwise method of variable selection. A separate analysis was done using total scores of anxiety and total scores of depression as dependent variables and demographic and COVID-19 variables as independent variables in the model.

Table 6 regression model data provides R^2^ = 0.39. The statistically significant predictors of anxiety scores were as follows: COVID-19 death in the family (β-coefficient = −11.68, *p* < 0.001, 95% CI = −14.09 to −9.27); COVID-19 cases in the family (β-coefficient = −2.21, *p* < 0.001, 95% CI = −3.18 to −1.23); female gender (β-coefficient = 3.25, *p* < 0.001, 95% CI = 2.53 to 3.98); and lower income group (β-coefficient = −1.12, *p* < 0.001, 95% CI = −1.58, 4.09 to −0.65). The predicted model for anxiety was as follows:

Anxiety scores = 29.32 + 3.25 gender (1 = male, 2 = female)—1.12 income group—2.2 COVID cases (1 = yes, 2 = no)—11.68 COVID death (1 = yes, 2 = no). Age was included in the model selection but was found not statistically significant for predicting anxiety (*p* = 0.98).

The regression analysis for depression showed R^2^ = 0.50. The statistically significant predictors of depression scores were as follows: COVID-19 death in the family (β-coefficient = −14.88, *p* < 0.001, 95% CI = −17.22 to −12.53); COVID-19 cases in the family (β-coefficient = −5.04, *p* < 0.001, 95% CI = −5.98 to −4.10); female gender (β-coefficient = 2.38, *p* < 0.001, 95% CI = 1.68 to 3.08); and lower income group (β-coefficient = −1.19, *p* < 0.001, 95% CI = −1.65 to −0.73) (Table 7). The predicted model for depression was as follows:

Depression scores = 47.07 + 2.38 gender (1 = male, 2 = female)—0.35 age—1.19 income group—5.04 COVID cases (1 = yes, 2 = no)—14.88 COVID death (1 = yes, 2 = no).

## 4. Discussion

### 4.1. Prevalence of Anxiety and Depression in Adolescents

In this study, the prevalence of major mental health conditions, anxiety, and depression, was 35.7% and 30.0%, respectively. Among the categories of severity for anxiety, 13.4% had mild, 19.1% had moderate, and 3.1% had severe symptoms. For depression, 15.1% had mild, 12.6% had moderate, 2.0% had moderate to severe symptoms, and 0.3% had severe symptoms. Overall, females had significantly higher scores for both anxiety and depression compared to their male counterparts (*p* < 0.001).

Our study showed a relatively higher prevalence of anxiety (36% vs. 20%) and depression (30% vs. 16%) when compared with another study conducted in the Kashmir valley of India [25]. The two studies were approximately similar in the following characteristics: both studies addressed school-going adolescents; males slightly predominated the females; both studies were cross-sectional in nature; both studies used the same data collection tools (GAD for anxiety and PHQ for depression), and females had a higher rate of anxiety and depression in both the studies. However, we observed much higher rates of anxiety and depression compared with the Indian Kashmir Valley study. This difference in the rates could be because the studies were conducted at two-time points of the COVID-19 pandemic—the Kashmir Valley study was conducted in 2021, whereas our study was conducted two years later when there were many more cases and deaths, as expected. As a result, we also anticipate an increase in the rate of adolescents’ mental illnesses among families with COVID-19 cases and/or deaths.

Another meta-analysis reviewed data from 18 studies on COVID-related lockdown and stay-at-home measures on the mental health of children and adolescents. In the later study, the prevalence of anxiety and depression in the Southeast Asia Region was 22% and 23%, respectively [26]. A Korean study [27] among school-going adolescents during the pandemic reported the overall prevalence of depression to be 19.8% which was slightly less when compared to our study. The prevalence of severe depression in the study from Korea was also lower (1.2% vs. 2.3%) when compared to the current study. Another meta-analysis [28] included 89 studies comprising 1,441,828 subjects of varying age groups and geographic regions (in developing and developed countries). The pooled prevalence of anxiety and depression symptoms during the pandemic was 32% and 33%, respectively, which were comparable to the current study. Another Chinese study showed a concurrence of anxiety and depression prevalences with our study results among adolescents [29].

The observed differences among many studies could be due to variations in study populations, geographic locations, and study designs. However, we used robust methods to analyze adolescents with and without COVID-19 cases and COVID-19 deaths in their families. In both univariate and multivariate analyses, our data convincingly quantified significantly higher scores of anxiety and depression (*p* < 0.001) among adolescents having COVID-19 cases or COVID-19 deaths in the family. Our community-based approach in rural West Bengal and the study methodologies were novel.

### 4.2. Gender Differences

We observed a significantly higher prevalence of both anxiety and depression among females in this study. Consistent with our findings, in a study among Chinese students aged 12–18 years during the COVID-19 pandemic, the prevalences for anxiety and depression were significantly higher in females than males (38.4% vs. 36.2%, *p* = 0.038, respectively for anxiety; and 45.5% vs. 41.7%, *p* = 0.001, respectively for depression) [29]. In the multivariate logistic regression model, females had a higher risk for anxiety (OR_anxiety_ = 1.64, 95% CI, 1.39 to 1.93) and depression (OR_depression_ = 1.15, 95% CI, 1.05 to 1.26). In another study conducted in 15 countries in Africa, the Americas, Asia, Europe, the Middle East, and the Pacific under the World Health Organization World Mental (WMH) Survey Initiative, women had significantly higher lifetime risk for anxiety-mood disorders and major depressive disorder than men [30]. In another prospective study in the U.S. [31], 113 middle school students, aged 11 to 14 years were assessed for three dimensions of anxiety (worry and oversensitivity, social concerns and concentration, and physiological anxiety) and depressive symptoms at an initial assessment and 1 year later. The U.S. study showed that anxiety symptoms (worry and oversensitivity in particular) predicted later depression symptoms more strongly for girls than for boys [31]. In our study, anxiety and depression often co-existed in adolescents. Further studies are needed to explore whether anxiety leads to depression, as observed in the U.S. study mentioned earlier.

### 4.3. Determinants of Anxiety and Depression

In the current study, GAD scores for anxiety and PHQ scores for depression were significantly higher (*p* < 0.001) among adolescents who had a family member with COVID-19 infection or death due to COVID-19 in the family—these findings were reiterated by the findings of Panda et al. (2020) from India [32]. Another meta-analysis by Lee et al. (2021) in Korea [27] found significantly higher prevalence of anxiety and depression in students who had a confirmed COVID-19 diagnosis compared to those who did not. Among other risk factors, one study reported a poor parent-child relationship associated with high anxiety and depression levels in adolescents during the pandemic [33]. However, our study was conducted primarily in a traditional rural Indian community and the occurrences of parent-child conflict are less common in the society. Lack of physical activities, online mode of education, and lack of interpersonal interactions were found as associated risk factors in several studies [27,34,35]; however, these data were not collected in our study.

While our study identified several important determinants of anxiety and depression in adolescents during the COVID-19 pandemic, it is important to note some limitations of our research.

### 4.4. Limitations and Strengths

This study has the following limitations that should be considered when interpreting the results: (1) Cross-sectional design: Due to the cross-sectional nature of the study, we cannot establish a causal relationship between COVID-19 and mental health problems in adolescents. (2) Single time point data collection: As data were collected at a single point in time, our study does not account for changes in mental health over the course of the pandemic or how its impact may have evolved. (3) Limited factors analyzed: We lack data on several potentially influential factors, such as access to recreational facilities, levels of physical activity, and parental education, which might affect adolescent mental health.

Despite these limitations, our study has notable strengths: (1) Community-based approach: In contrast to many previous institute-based studies, our research involved a rural community in West Bengal, providing insights into a less-studied population. (2) Random selection: Households were selected using a random selection procedure, enhancing the representativeness of our sample and reducing potential selection bias. (3) Direct data collection: Data were collected and monitored through house-to-house visits and in-person interviews, ensuring accuracy and potentially reducing the impact of confounding variables.

These strengths contribute to the validity and uniqueness of our findings while acknowledging the constraints inherent to the study design.

## 5. Conclusions and Recommendations

This study found that the prevalence of depression and anxiety was significantly higher among adolescents from families with COVID-19 cases and COVID-19 deaths. These findings underscore the need for immediate, targeted approaches to support adolescent mental health and stress-coping strategies. Families, schools, and healthcare providers have specific roles to play such as the following:Families: Encourage open communication to facilitate healthy decision-making and provide supervision to prevent antisocial activities.Schools: Develop and implement specific training modules to help students cope with emergencies and their aftermath. Train teachers and staff to build a safe and supportive environment.Healthcare providers: Clinicians and nurses can provide coaching and counseling and educate students on self-care and coping skills. Community-based health workers should be trained to provide early interventions for adolescent mental health and encourage positive parenting practices.

As pandemics cause unprecedented changes involving health, social, financial, and many other dimensions, a holistic approach is crucial. This approach should involve all stakeholders, including young people, schoolteachers, parents, clinicians, social workers, policymakers, and the media, to ensure the mental well-being of adolescents during and after a pandemic.

## Figures and Tables

**Figure 1 diseases-12-00233-f001:**
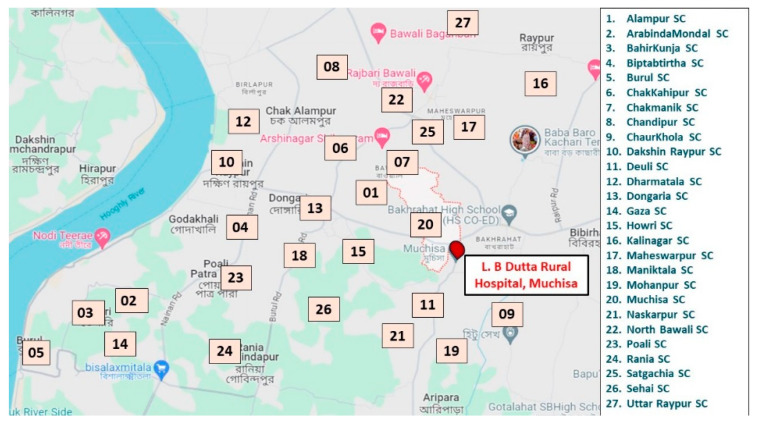
The study areas and locations of the 27 subcenters (marked in boxes) and a local rural hospital (L.B. Dutta Rural Hospital) in *Muchisa*, West Bengal, India.

**Figure 2 diseases-12-00233-f002:**
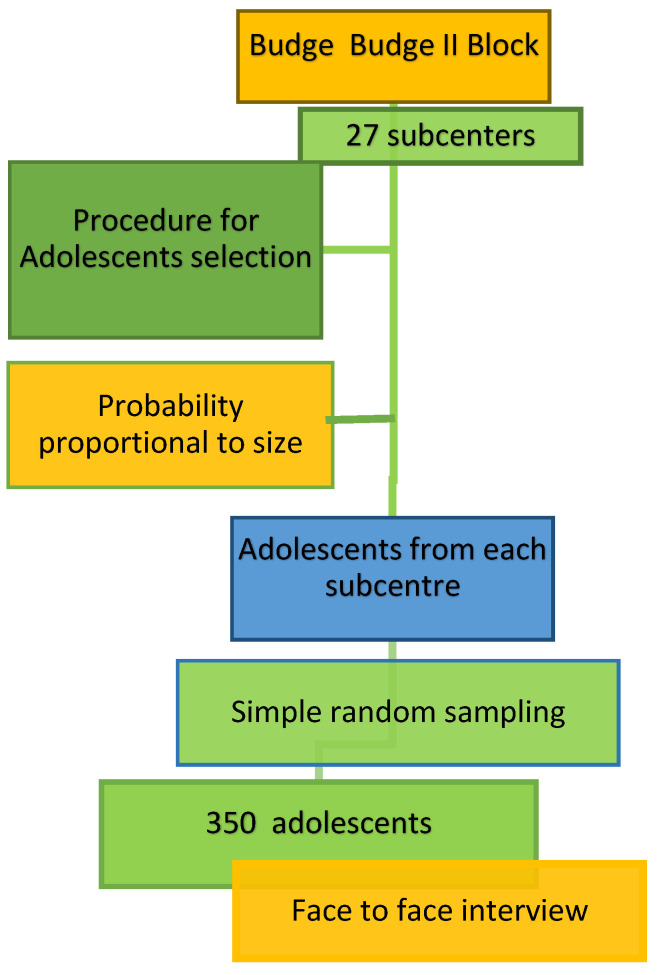
Flow diagram showing the selection of participants.

**Figure 3 diseases-12-00233-f003:**
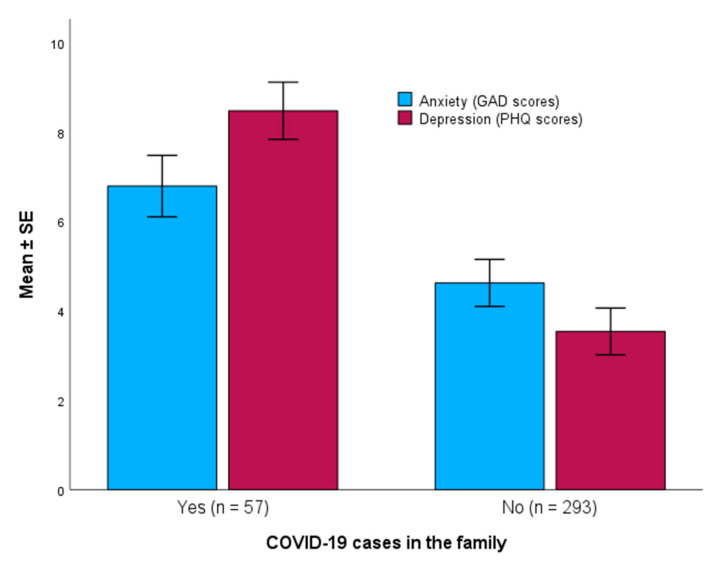
Scores of anxiety and depression (mean ± *SE*) among adolescents who had COVID-19 cases in the family (*p* < 0.001, Mann-Whitney U test).

**Figure 4 diseases-12-00233-f004:**
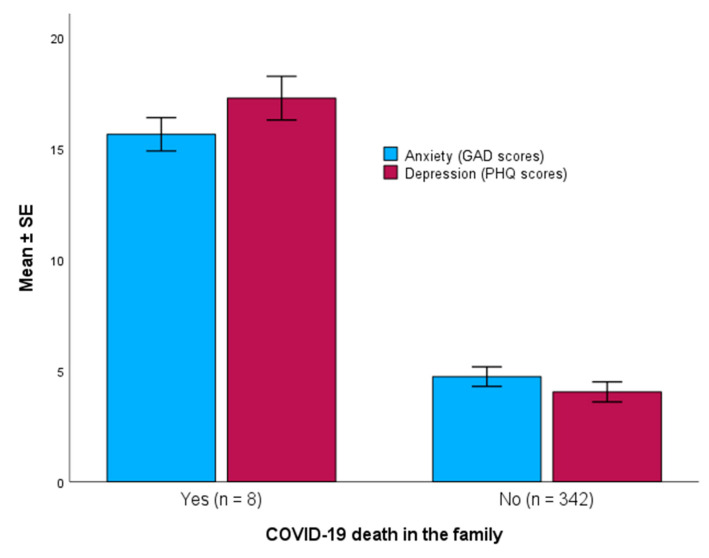
Scores of anxiety and depression (mean ± *SE*) among adolescents who had COVID-19 death in the family (*p* < 0.001, Mann-Whitney U test).

**Table 1 diseases-12-00233-t001:** Demographic and health information of the participants (*n* = 350).

Variable	Mean ± *SD*	No (%)
Age (year)	15.59 ± 1.93	
Gender:		
Male		182 (52)
Female		168 (48)
Number of family members, mean ± *SD* (range)	5.0 ± 0.7 (4–7)	
Family income (Rupees) (1 US $ = 83.95 Rupees; *n* = 349)		
<25,000		136 (39.0)
25,000–50,000		156 (44.7)
50,001–90,000		46 (13.2)
≥90,001		11 (3.2)
Smoking habit of the participant		
Current smoker		9 (2.6)
Occasional		12 (3.4)
Never smoked		329 (94.0)
Family history of COVID-19		
Did anybody have COVID-19 (Yes)		57 (16.3)
Did anybody die of COVID-19 in your family (Yes)		8 (2.3)
Did you suffer from COVID-19 (Yes)		5 (1.4)
Are you currently having any symptoms (Yes)		27 (7.7)
Do you have symptoms of anxiety or depression (such as feeling sad or anxious, feeling easily frustrated or restless, having trouble sleeping, or others)		3 (0.9)
Who did you consult for your current illnesses (*n* = 27)		
Qualified medical doctor		4 (14.8)
Pharmacy medicine seller		16 (59.3)
Self-medication		4 (14.8)
None		3 (11.1)
How do you rate your current health (*n* = 326)		
Very good		287 (88.0)
Good or moderate		31 (9.5)
Bad or very bad		8 (2.5)

**Table 2 diseases-12-00233-t002:** Prevalence of anxiety categories of severity by gender.

GAD Categories	Total	Male	Female	*p*-Value ^1^
Minimal (0–4)	225 (64.3)	145 (79.7)	80 (47.6)	<0.001
Mild (5–9)	47 (13.4)	28 (15.4)	19 (11.3)	
Moderate (10–14)	67 (19.1)	3 (1.6)	64 (38.1)	
Severe (≥15)	11 (3.1)	6 (3.3)	5 (3.0)	
Total	350 (100)	182 (100)	168 (100)	

^1^ Chi-square test.

**Table 3 diseases-12-00233-t003:** Prevalence of depression categories of severity by gender.

PHQ Categories	Total	Male	Female	*p*-Value ^1^
None to Minimal (0–4)	245 (70.0)	145 (79.7)	100 (59.5)	<0.001
Mild (5–9)	53 (15.1)	28 (15.4)	25 (14.9)	
Moderate (10–14)	44 (12.6)	3 (1.6)	41 (24.4)	
Moderately Severe (15–19)	7 (2.0)	5 (2.7)	2 (1.2)	
Severe (20–27)	1 (0.3)	1 (0.5)	0 (0)	
Total	350 (100)	182 (100)	168 (100)	

^1^ Chi-square test.

**Table 4 diseases-12-00233-t004:** Gender difference in anxiety and depression scores.

Variable	Male (*n* = 182)	Female (*n* = 168)	*p*-Value ^1^
Anxiety score: Median (IQR)	3 (2–4)	5 (3–12)	<0.001
Depression score: Median (IQR)	2 (1–4)	4 (1–10)	<0.001

^1^ Mann-Whitney U test; IQR = interquartile range.

**Table 5 diseases-12-00233-t005:** Bivariate correlation between variables.

		Age	Gender	Income Group	COVID Cases	COVID Death	Depression	Anxiety
Age	*r*	1	0.064	−0.224	0.015	−0.142	−0.022	0.123
	*p*		0.231	<0.001	0.781	0.008	0.677	0.022
	*n*		350	349	350	350	350	350
Gender (male = 1; female = 2)	*r*		1	−0.127	−0.010	0.070	0.246	0.376
	*p*			0.018	0.853	0.189	<0.001	<0.001
	*n*			349	350	350	350	350
Income group	*r*			1	0.098	0.060	−0.272	−0.292
	*p*				0.067	0.265	<0.001	<0.001
	*n*				349	349	349	349
COVID cases	*r*				1	−0.067	−0.399	−0.185
	*p*					0.208	<0.001	<0.001
	*n*					350	350	350
COVID death	*r*					1	−0.432	−0.377
	*p*						<0.001	<0.001
	*n*						350	350
Depression	*r*						1	0.856
	*p*							<0.001
	*n*							350
Anxiety								1

**Table 6 diseases-12-00233-t006:** Multiple linear regression analysis to predict anxiety.

Independent Variable	*β*-Coefficient	*SE*	*p*-Value	95% Confidence Intervals	
				Lower	Upper
(Constant)	29.32	2.68			
COVID death in the family (1 = yes, 2 = no)	−11.68	1.23	<0.001	−14.09	−9.27
COVID cases in the family (1 = yes, 2 = no)	−2.21	0.50	<0.001	−3.18	−1.23
Gender (1 = male, 2 = female)	3.25	0.37	<0.001	2.53	3.98
Income groups (Rupee) ^1^	−1.12	0.24	<0.001	−1.58	−0.65

^1^ Income groups: 1 = <25,000, 2 = 25,000–50,000, 3 = 50,001–90,000, 4 = ≥90,001 Rupees.

**Table 7 diseases-12-00233-t007:** Multiple linear regression analysis to predict depression.

Independent Variable	*β*-Coefficient	*SE*	*p*-Value	95% Confidence Intervals	
				Lower	Upper
(Constant)	47.07	3.16			
COVID death in the family (1 = yes, 2 = no)	−14.88	1.19	<0.001	−17.22	−12.53
COVID cases in the family (1 = yes, 2 = no)	−5.04	0.48	<0.001	−5.98	−4.10
Gender (1 = male, 2 = female)	2.38	0.36	<0.001	1.68	3.08
Income groups (Rupee) ^1^	−1.19	0.23	<0.001	−1.65	−0.73
Age	−0.35	0.09	<0.001	−0.53	−0.16

^1^ Income groups: 1 = <25,000, 2 = 25,000–50,000, 3 = 50,001–90,000, 4 = ≥90,001 Rupees.

## Data Availability

The data collection questionnaires are provided in Appendix A and Appendix A. The data presented in this study are available on request from the corresponding author.

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
