# Peer review of "COVID-19 Pandemic Increases the Risk of Anxiety and Depression among Adolescents: A Cross-Sectional Study in Rural West Bengal, India"

_diseases, 2024, doi:10.3390/diseases12100233_

Round 1

Reviewer 1 Report

Comments and Suggestions for Authors

Well written manuscript. Two suggestions on the statistical analysis part:

1) In Table 3, Mann-Whitney U test is used to compare mean (SD) of depression score vs. gender, as well as, anxiety score vs. gender. However, this non-parametric test is comparing median. Suggest to report the median (IQR) values or use one-way ANOVA (Type III) to maintain mean (SD) values if assumption allows.

2) The researchers have developed prediction models, even though it is not part of the study aim. In Table 5, a prediction model was formulated to predict anxiety score while in Table 6, a prediction model was generated to predict depression score. It would be beneficial to reader if these models were assessed for sensitivity and specificity statistics. Why would people use the predictive models over the well established questionnaires (GAD and PHQ9) assess their mental health?

Overall, results were correctly interpreted but need to update the statistical steps. Discussion points were relevant and good recommendations were outlined.

Comments on the Quality of English Language

Good English proficiency, the report can be read easily. 

Reviewer 2 Report

Comments and Suggestions for Authors

Dear Editor and Authors,

I have thoroughly reviewed the manuscript "COVID-19 Pandemic Increases the Risk of Anxiety and Depression among Adolescents: A Cross-Sectional Study in Rural West Bengal, India". While the study investigates an important and timely topic, several areas raise concerns about the quality and rigor of the research. I recommend addressing the following points to strengthen the manuscript:

1. Rationale for focusing on underprivileged rural communities: The authors' choice to focus on underprivileged rural communities is understandable given the likely poor healthcare access in these areas. However, this alone does not provide sufficient rationale for the study. The introduction lacks a comprehensive literature review on this specific topic. To address this: Provide a thorough review of relevant studies examining COVID-19's impact on adolescent mental health in rural or underprivileged areas; Discuss any existing research on mental health outcomes in rural West Bengal or similar regions in India; Highlight how your study builds upon or differs from previous research in this area.

2.Justification and research motivation: The manuscript currently lacks a strong justification for conducting this particular study. To improve this: Clearly articulate the knowledge gap your research aims to address; Explain why studying adolescent mental health in rural West Bengal during the COVID-19 pandemic is significant; Discuss potential implications of your research for public health policy or interventions in similar settings; Consider framing your study within a broader context of global adolescent mental health research during the pandemic.

3.Sample size calculation: The method used to calculate the sample size is not adequately supported. To address this: Provide citations for the formula used in your sample size calculation; Explain the parameters used in the calculation (e.g., expected prevalence, confidence level, margin of error); Justify why these particular parameters were chosen for your study.

4.Random sampling procedure: The current description of your sampling method does not clearly demonstrate a truly random sampling procedure. To improve this: Provide a step-by-step explanation of your sampling process; If a stratified or cluster sampling method was used, clearly describe the strata or clusters and how they were selected; Explain any potential biases in your sampling method and how you attempted to mitigate them; If true random sampling was not feasible, acknowledge this limitation and discuss its potential impact on the generalizability of your results.

5.Hypotheses and statistical analyses: The manuscript would benefit from more clearly specified hypotheses and justifications for the statistical tests used.

In conclusion, while this study addresses an important topic, significant revisions are necessary to improve its scientific rigor and clarity. Addressing these points will strengthen the manuscript and enhance its contribution to the field of adolescent mental health research in the context of the COVID-19 pandemic.

Reviewer 3 Report

Comments and Suggestions for Authors

Thank you for addressing this important issue. However, there are very few minor corrections that need to be clarified. 

Introduction

1. Lines 76 and 77; how do you decide that the prevalence of depression and anxiety were related to COVID-19. This is an argued note and should be clarified. 

Methods

2. Regarding sample size calculations: what the role of adjusting for non-responses (15%) in the equation? I tried many times to do the calculations, bur did not get your result (313). 

3. In line 143; what was the result of translation and back translation, have you done any changes?

4. Need to describe the scoring system for both GAD-7 and PHQ-9?

5. In line 155, do you have the same cut off points for both scales? if yes, what is the reference for that. In literature, there are different categorization for those two scales. 

6. In line 176; (only 3 adolescence.....). this sentence is confusing as it  does not appear in the table. 

7. In line 246, age was not a significant predictor, therefore it it was excluded? It is not clear if that (age) entered the regression in both analyses or just in one analysis. 

8. In lines 256-257; age mentioned twice. 

Discussion

9.  In line 270; can you compare your results with Kasmir valley study? If yes, are the characters of the population is similar or approximately similar?

Round 2

Reviewer 2 Report

Comments and Suggestions for Authors

I recommend publishing the current version.